# The Effects of Dietary Silybin Supplementation on the Growth Performance and Regulation of Intestinal Oxidative Injury and Microflora Dysbiosis in Weaned Piglets

**DOI:** 10.3390/antiox12111975

**Published:** 2023-11-07

**Authors:** Long Cai, Ge Gao, Chenggang Yin, Rong Bai, Yanpin Li, Wenjuan Sun, Yu Pi, Xianren Jiang, Xilong Li

**Affiliations:** Key Laboratory of Feed Biotechnology of the Ministry of Agriculture and Rural Affairs, Institute of Feed Research, Chinese Academy of Agricultural Sciences, Beijing 100081, China; 82101211211@caas.cn (L.C.); 82101192355@caas.cn (G.G.); 82101231287@caas.cn (C.Y.); rong.bai@wur.nl (R.B.); sunwenjuan@caas.cn (W.S.); piyu@caas.cn (Y.P.); jiangxianren@caas.cn (X.J.)

**Keywords:** silybin, growth performance, intestinal health, microflora dysbiosis, mitochondria function, weaned piglet

## Abstract

Oxidative stress is the major incentive for intestinal dysfunction in weaned piglets, which usually leads to growth retardation or even death. Silybin has caught extensive attention due to its antioxidant properties. Herein, we investigated the effect of dietary silybin supplementation on growth performance and determined its protective effect on paraquat (PQ)-induced intestinal oxidative damage and microflora dysbiosis in weaned piglets. In trial 1, a total of one hundred twenty healthy weaned piglets were randomly assigned into five treatments with six replicate pens per treatment and four piglets per pen, where they were fed basal diets supplemented with silybin at 0, 50, 100, 200, or 400 mg/kg for 42 days. In trial 2, a total of 24 piglets were randomly allocated to two dietary treatments with 12 replicates per treatment and 1 piglet per pen: a basal diet or adding 400 mg/kg silybin to a basal diet. One-half piglets in each treatment were given an intraperitoneal injection of paraquat (4 mg/kg of body weight) or sterile saline on day 18. All piglets were euthanized on day 21 for sample collection. The results showed that dietary supplementation with 400 mg/kg silybin resulted in a lower feed conversion ratio, diarrhea incidence, and greater antioxidant capacity in weaned piglets. Dietary silybin enhanced intestinal antioxidant capacity and mitochondrial function in oxidative stress piglets induced by PQ. Silybin inhibited mitochondria-associated endogenous apoptotic procedures and then improved the intestinal barrier function and morphology of PQ-challenged piglets. Moreover, silybin improved intestinal microbiota dysbiosis induced by the PQ challenge by enriching short-chain fatty-acid-producing bacteria, which augmented the production of acetate and propionate. Collectively, these findings indicated that dietary silybin supplementation linearly decreased feed conversion ratio and reduced diarrhea incidence in normal conditions, and effectively alleviated oxidative stress-induced mitochondrial dysfunction, intestinal damage, and microflora dysbiosis in weaned piglets.

## 1. Introduction

The intestine is the main organ for digestion, absorption, and metabolism of nutrients and plays a critical role in preventing harmful substances from entering the circulatory system [1,2]. Oxidative stress usually induces intestinal injury and microflora dysbiosis, subsequently damaging the physiological function of intestinal epithelial cells, ultimately resulting in diarrhea, growth retardation, and even death [1,3,4]. The situation becomes worse for piglets during weaning due to their immature intestines. Therefore, maintaining proper redox balance through nutritional intervention may help to alleviate intestinal oxidative injury, thereby improving intestinal health and growth performance in weaned piglets.

Currently, research on improving intestinal health through dietary antioxidants, including flavonoid supplementation, has been gradually emphasized. Silymarin is a kind of flavonoid lignans isolated from milk thistle seeds and contains silybin (A/B), isosilybin (A/B), silychristin, and silydianin, which has been used for a long time as a nutraceutical and food additive for the maintenance of systemic health [5]. Silybin is a quasi-equimolar mixture of two diastereomers named silybin A and silybin B, which is a major bioactive molecule with various biological properties, including antioxidant, anticancer, and immunomodulatory [6,7]. Emerging studies showed that dietary silymarin/silybin exhibited health-promoting effects for animals [8,9]. For instance, diets supplemented with silymarin could improve growth performance, intestinal health, and egg quality in poultry [9,10]. Dietary supplementation with silymarin (40 g/day) during transition and lactation enhanced the serum antioxidant activity of sows [11]. In addition, dietary micelle silymarin treatment improved the growth performance, nutrient digestibility, and meat quality of finishing pigs [12]. Interestingly, previous studies also found that silybin could upregulate the relative abundance of beneficial bacteria and decrease the relative abundance of fecal pathogenic microbes [13,14]. However, to the best of our knowledge, any beneficial effects and the underlying mechanism of silybin on growth performance and intestinal health in weaned piglets remain elusive.

Thus, this study aimed to investigate the effect of dietary silybin supplementation on growth performance and subsequently determined whether dietary silybin could alleviate intestinal oxidative injury and microflora dysbiosis caused by paraquat (PQ) in weaned piglets. PQ is a well-known electron acceptor that can induce the body to produce excessive reactive oxygen species (ROS), widely used to structure an oxidative stress model [15,16]. To determine the appropriate dosage of silybin, we evaluated the growth performance, diarrhea incidence, and plasma antioxidant enzyme activity of piglets fed with four amounts of silybin supplementation, namely 50, 100, 200, or 400 mg/kg. Then, we examined the effects of dietary proper silybin dose on antioxidant capacity, mitochondrial function, intestinal barrier, and gut microbiota in PQ-challenged piglets to reveal the potential mechanism of silybin in regulating intestinal health.

## 2. Materials and Methods

### 2.1. Animal Ethics Statement

The animal protocol in this study was approved by the Animal Care and Use Committee of the Institute of Feed Research of the Chinese Academy of Agricultural Sciences (IFR-CAAS20220228, IFR-CAAS20220428). All animal experiments were performed following the ARRIVE guidelines [17].

### 2.2. Animals and Experimental Design

In the first trial, a total of 120 healthy weaned piglets (Duroc × Landrace × Yorkshire) with an age of 28 ± 2 d and an initial body weight of 8.28 ± 0.48 kg were randomly assigned into 5 treatments with 6 replicate pens per treatment and 4 piglets per pen (2 barrows and 2 females). The treatment diets were basal diets supplemented with 0, 50, 100, 200, or 400 mg/kg of silybin, respectively. The silybin (purity > 97%) used in this trial was bought from Panjin Tianyuan Pharmaceutical Co., Ltd. (Panjin, China). This trial lasted for 42 days.

In the second trial, a total of 24 healthy weanling crossbred piglets (Duroc × Landrace × Yorkshire, with an age of 26 ± 1 d and an initial body weight of 7.68 ± 0.37 kg) were randomly allocated to 2 dietary treatments with 12 replicates (pens) per treatment and 1 piglet per pen according to initial body weight and sex (same male and female): a basal diet or adding 400 mg/kg silybin to a basal diet. One-half piglets in each treatment were given an intraperitoneal injection of paraquat (PQ) (methyl viologen hydrate, Huaxia Chemical Reagent Co., Ltd., Chengdu, China) at the dose of 4 mg/kg body weight in saline on the 18th day of the experiment, and the others were intraperitoneally injected with the same volume of sterile saline solution (Figure 1A). The dose of PQ was selected in the current study according to a previous report [18]. The amount of silybin in the second trial was determined according to the results of the first trial. This trial lasted for 21 days.

Nutrient levels of the corn–soybean meal basal diet are formulated based on the National Research Council (2012) nutritional requirements (Appendix A) and do not contain any antibiotic growth promoters. The temperature of the nursery house was controlled at 26–28 °C, and the relative humidity was kept at 55–65%. Piglets were given ad libitum access to feed and fresh water through a feed trough and nipples in pens with slatted floors.

### 2.3. Growth Performance and Diarrhea Incidence

In trial 1, body weight and feed intake were recorded on days 0, 14, 28, and 42 in each pen to evaluate the average daily gain (ADG), average daily feed intake (ADFI), and feed conversion ratio (FCR). In trial 2, body weight and feed intake were recorded on days 0, 18, and 21 to evaluate the growth performance. The fecal score was recorded once every morning via visual inspection according to the five-point stool consistency scoring system: 1 = hard, dry pellet; 2 = firm, formed stool; 3 = soft, moist stool that maintains its shape; 4 = soft, shapeless feces; and 5 = flowable liquid. A score exceeding 3 indicated diarrhea, and the diarrhea incidence (%) was calculated as a percentage of the count of piglets with diarrhea divided by the total number of piglets in each treatment.

### 2.4. Sample Collection

On the morning of days 14 and 42 of trial 1, one piglet per pen was randomly selected for blood sample collection via venipuncture from the jugular vein of the piglets. In trial 2, the blood samples of all the piglets were collected on the morning of day 21. The blood samples were centrifuged at 3000× *g* for 10 min at 4 °C to obtain plasma that was stored at −20 °C for further analyses. At the end of trial 2, the piglets were stunned using a portable electrical stunner (the output voltage is 220 V) in a lidless 100 × 65 × 54 cm plastic box and bled quickly to be euthanized. The abdomen was incised longitudinally to collect the tissues. Then, the partial small intestine tissues were fixed in fresh 4% paraformaldehyde for further analysis. Samples of jejunum mucosa were collected and placed into cryogenic vials (Corning Incorporated, New York NY, USA), frozen in liquid nitrogen, and stored at −80 °C for analysis. In addition, cecal digesta was collected and stored at −80 °C for microbiome and short-chain fatty acid (SCFAs) determination.

### 2.5. Assay of Antioxidant Indices in Plasma and Intestinal Mucosa

The activities of superoxide dismutase (SOD), catalase (CAT), and glutathione peroxidase (GSH-Px), and the level of malondialdehyde (MDA) and hydrogen peroxide (H_2_O_2_) in the plasma and jejunal mucosa were gauged with commercial assay kits as instructed (Jiancheng Bioengineering Institute, Nanjing, China). The absorption peaks of CAT, SOD, GSH-Px, MDA, and H_2_O_2_ were 405 nm, 450 nm, 412 nm, 532 nm, and 405 nm, respectively. The test step of the above kits follows the manufacturer’s instructions.

### 2.6. The Activities of Mitochondrial Complex and ATP Content Assay

The activities of mitochondrial complex I and V in jejunal mucosa were gauged with commercially available kits (Beijing Solarbio Science & Technology Co., Ltd., Beijing, China) according to the manufacturer’s instructions. The ATP contents were detected using a corresponding assay kit (Nanjing Jiancheng Bioengineering Institute, Nanjing, China) according to the manufacturer’s protocols.

### 2.7. Caspase 3 and Caspase 9 Activity

The activities of caspase 3 and caspase 9 in jejunal mucosa were performed using a specific reagent kit (Beyotime Biochem. Co., Ltd., Shanghai, China) as described by the manufacturer’s instructions. Briefly, the intestinal tissue was homogenized in the tissue grinder and centrifuged at 4 °C for 12 min to obtain the supernatant. Subsequently, the supernatant was incubated with Ac-DEVD-pNA (for caspase 3) or Ac-LEHD-pNA (for caspase 9) at 37 °C for 60 min. Then, the absorbance at 405 nm was measured employing a microplate reader (BioTek Instruments, Inc., Winooski, VT, USA).

### 2.8. Detection of Biomarkers of Intestinal Barrier Dysfunction

The plasma contents of D-lactate and diamine oxidase (DAO) activity in piglets were determined using commercial kits as per the protocol provided by the manufacturer (Jiancheng Bioengineering Institute, Nanjing, China).

### 2.9. Mucin 2 Content Assay

The contents of mucin 2 (MUC 2) in the jejunum were measured using enzyme-linked immunosorbent assay (ELISA) kits (Jiangsu Meimian Industrial Co., Ltd., Yancheng, China) as described by the manufacturer’s instructions.

### 2.10. Histopathological Staining

The small intestinal tissues were embedded in paraffin after fixation with 4% paraformaldehyde at ambient temperature overnight. Sectioned at 5 μm slices were prepared using a pathology slicer (Leica RM2016, Nussloch, Germany), which were then colored with hematoxylin/eosin (H&E) for small intestinal morphological observation. In addition, paraffin sections of the jejunum were stained with alcian blue/periodic acid–Schiff (AB-PAS) to check for cupped cells. The slices were observed and photographed under an upright optical microscope (Niko, Tokyo, Japan).

### 2.11. Real-Time Quantitative PCR Analysis (qPCR)

RNA extraction and qPCR were performed according to the procedure described in a previous study [19]. In brief, total RNA was extracted from intestinal mucosa samples using the Trizol reagent (Thermo Fisher Scientific, Inc., Boston, MA, USA) according to the protocol provided by the manufacturer. Then, reverse transcription was performed to obtain complementary deoxyribonucleic acid (cDNA) after the determination of the concentration and quality of RNA. qPCR analysis was processed on a CFX96 Touch real-time PCR instrument (Bio-Rad Laboratories Inc., Berkeley, CA, USA). The quantification of target gene relative expression was calculated using the 2^−ΔΔCT^ method. The glyceraldehyde-3-phosphate dehydrogenase (GAPDH) served as a housekeeping gene. The primers used in the qPCR assay are presented in Appendix A.

### 2.12. Western Blotting Analysis

Sample preparation and Western blot analysis were processed as described previously [20]. Briefly, jejunal mucosa was pulverized and lysed in RIPA buffer containing 1% phosphatase and protease inhibitors (Huaxing Biotechnology, Beijing, China). The protein concentration of the lysate was assayed using the BCA method after centrifugation. The proteins were separated with SDS-polyacrylamide gel electrophoresis gels and then transferred to PVDF membranes (Bio-Rad Laboratories Inc., Berkeley, CA, USA) using the wet method, followed by blocking with 5% BSA. Then, the membranes were incubated with the primary antibody at 4 °C overnight with gentle shaking, washed with tris buffer saline added with 0.1% Tween 20, and subsequently incubated for another 3 h at normal temperature with the specific secondary antibody. The membranes were visualized using the Western Bright ECL Kit (Huaxing Biotechnology, Beijing, China) and placed into the Imaging System (Bio-Rad Laboratories, Inc., Berkeley, CA, USA). Quantification was performed with the Image J analyzer software 1.51 (National Institute of Health, Bethesda, MD, USA), and the level of protein expression was normalized to GAPDH. All antibody information is included in Appendix A.

### 2.13. Gut Microbiome Analysis

The genomic DNA of the microbial community was collected from the cecal chyme using a commercial kit (Omega Bio-Tek, Norcross, GA, USA) as per the protocol of the manufacturer. The specific primer pairs were used to amplify the hypervariable regions V3-V4 of the bacterial 16S rRNA gene. The PCR amplification products were quantified after being purified using the commercial kit (Axygen Biosciences, Union City, CA, USA). Sequencing of qualified libraries was then performed on a specific platform (Illumina, San Diego, CA, USA). The reads generated an amplicon sequence variant (ASV) after denoising using DADA2 under the default parameters [21]. The subsequent data analysis was performed using the online platform, Majorbio Cloud Platform (https://cloud.majorbio.com/ (accessed on 21 February 2023)). The raw data were deposited in the NCBI Sequence Read Archive (SRA) database (Accession number: PRJNA933795).

### 2.14. Determinations of SCFAs in the Cecal Digesta

The concentrations of SCFAs were detected using ion chromatography. In brief, the chyme samples were dissolved in precooled dilution water (including ZnSO_4_∙7H_2_O and K_4_Fe (CN)_6_∙3H_2_O) and centrifuged at 4 °C for 10 min at 10,000 rpm. Afterward, they were filtered, and the supernatants were diluted with distilled water at a 1:4 ratio. Then, the supernatant was subjected to SCFA detection with an 883 Ion Chromatograph (IC; Metrohm, Switzerland).

### 2.15. Statistical Analysis

The data related to growth performance and plasma antioxidant capacity in trial 1 were analyzed with one-way ANOVA using the general linear model (GLM) procedure of SPSS 19 (IBM, Armonk, NY, USA). The dosage-related effect of supplemental silybin was computed with GLM using the contrast command for the linear and quadratic effects. The treatment comparisons were conducted using Tukey’s honest significant difference test for multiple testing, and the Chi-square test was used to analyze the incidence of diarrhea. In addition, one-way analysis of variance followed by Tukey’s honest significant difference test was used for data statistical analysis in trial 2 except for the gut microbiome data. The Wilcoxon test and Kruskal–Wallis test were used to compare the differences in microbial communities between two groups and multiple groups, respectively. The data are expressed as mean with standard error. A *p*-value < 0.05 was accepted as significant, and 0.05 ≤ *p*-value < 0.1 was accepted as a significant trend.

## 3. Results

### 3.1. Effects of Dietary Silybin Supplementation on Growth Performance, Diarrhea Incidence, and Antioxidant Capacity in Weaned Piglets (Trial 1)

As shown in Table 1, dietary supplementation with silybin had no effects on BW and ADFI during the trial time (*p* > 0.05). However, compared with the control group, supplementation with silybin at 400 mg/kg significantly decreased FCR during days 28–42 (*p* < 0.05). In addition, a linear increase was observed in ADG on days 28–42 in response to dietary silybin supplementation (*p* < 0.05), while linear decreases were observed in FCR on days 28–42 and days 0–42 in response to dietary silybin supplementation (*p* < 0.05). As to diarrhea incidence, compared with the control group, dietary supplementation with 400 mg/kg silybin significantly decreased the incidence of diarrhea in weaned piglets (*p* < 0.05).

As shown in Table 2, there was no significant effect on plasma CAT and GSH-Px activities on day 14 or SOD activity on day 42 among all groups (*p* > 0.05). Compared with the control group, dietary supplementation with 400 mg/kg silybin significantly increased plasma SOD activity on day 14, CAT activity on day 42, and decreased MDA content on day 42 (*p* < 0.05). In addition, dietary supplementation with 200 mg/kg silybin tended to plasma reduce MDA content on day 14 and day 42 (*p* = 0.079 and *p* = 0.099, respectively). Both linear and quadratic effects were observed on plasma SOD activity and MDA content on day 14 in response to dietary silybin addition (*p* < 0.05). There was a linear response in the plasma activities of CAT and GSH-Px and the MDA content on day 42 with silybin supplementation (*p* < 0.05). According to the above results, dietary silybin at 400 mg/kg was the optimal supplementation amount to decrease diarrhea and improve antioxidant capacity and growth performance in weaned piglets.

### 3.2. Dietary Silybin Supplementation Alleviated the Redox Imbalance and Eliminated the Growth Retardation Induced by Paraquat in Piglets (Trial 2)

The effect of silybin on the growth performance and systemic antioxidant properties of weaned piglets was first examined. There was no difference in ADG, ADFI, and FCR between the control and the silybin-supplemented group before the piglets were challenged with PQ (*p* > 0.05) (Appendix A–C). However, dietary silybin supplementation increased the ADG of the weaned pigs after the PQ challenge (*p* < 0.05) (Figure 1B,D). Plasma redox balance was evaluated by detecting biomarkers of oxidative stress. Compared with the control group, PQ challenge triggered a strong systemic redox imbalance, evidenced by decreased GSH-Px activity and increased levels of MDA, while dietary silybin supplementation alleviated the decline in the activity of GSH-Px and markedly reduced MDA content induced by PQ challenge (*p* < 0.05). In addition, dietary silybin supplementation increased the activity of the CAT (*p* < 0.05), although no change was observed for SOD activity or H_2_O_2_ level (*p* > 0.05) (Figure 1E–I). Overall, these results indicated that dietary silybin supplementation relieved oxidative stress and improved the growth arrest induced by PQ in weaned piglets.

### 3.3. Dietary Silybin Administration Alleviated Paraquat-Induced Intestinal Redox Imbalance in Piglets (Trial 2)

We further studied the role of silybin on the intestinal antioxidant capacity of piglets. Compared to the control group, the activities of CAT and SOD in the jejunum were decreased in the PQ challenge group, while the MDA content was evidently increased (*p* < 0.05). However, the addition of silybin to the diet reversed the decline in CAT and SOD activity and the increase in MDA content induced by the PQ challenge (*p* < 0.05) (Figure 2A–D). qPCR results indicated that, compared to the PQ challenge group, dietary supplementation with silybin tended to upregulate the mRNA abundance of *SOD1* (*p* = 0.07) and *SOD2* (*p* = 0.05). However, there was no alteration observed in the expression of the *CAT*, *GPX1*, *GPX2*, and *GPX4* genes (*p* > 0.05) (Figure 2E). As shown in Figure 2F, PQ treatment significantly downregulated the mRNA abundance of *Nrf2* and *Keap1* in the jejunum of weaned piglets compared to the control group (*p* < 0.05). Intriguingly, dietary silybin supplementation recovered the expression of the *Nrf2* gene under the PQ challenge (*p* < 0.01). Collectively, these results showed that dietary silybin supplementation tremendously alleviates intestinal oxidative stress induced by PQ challenge in piglets.

### 3.4. Dietary Silybin Supplementation Protected against PQ-Induced Mitochondrial Injury (Trial 2)

As shown in Figure 3, PQ challenge significantly downregulated the mRNA abundance of the mitochondrial division gene dynamin 1 (*DNM1*) (*p* < 0.01), along with a downward trend in the mRNA expression of the fusion gene mitofusin 2 (*MFN2*) compared to those in the control (*p* = 0.08), while silybin supplementation partially improved mitochondrial biogenesis gene expression (Figure 3A–E). In addition, silybin addition abolished the PQ-induced decrease in NADH ubiquinone oxidoreductase core subunit V2 (*NDUFV2*) (*p* = 0.05) and ATP synthase (*ATP5H*) (*p* < 0.05) gene expression in the jejunum. Correspondingly, dietary silybin remarkably increased the activities of mitochondrial complex I and complex V, thereby enhancing ATP content in the jejunal mucosa of PQ-challenged piglets (*p* < 0.05) (Figure 3L–N). Moreover, Spearman’s correlation analysis found a significant correlation between mitochondrial function-related genes and Nrf2 signaling pathway genes (Figure 3K). These data confirmed that dietary silybin protected against PQ-induced mitochondrial dysfunction in the jejunum of piglets.

### 3.5. Dietary Silybin Addition Inhibited PQ-Induced Intestinal Apoptotic (Trial 2)

As shown in Figure 4A,B, the PQ challenge induced a significant increase in caspase 3 and caspase 9 activities in the jejunum. However, dietary silybin mitigated the increased activities of caspase 3 and caspase 9 in the jejunum of PQ-challenged piglets (*p* < 0.05). Consistent with these results, Western blot analysis showed dietary silybin addition significantly reduced the protein expression of Cleaved caspase-3 and Bcl-2-associated-X-protein (Bax) (*p* < 0.05) and significantly enhanced the protein expression of B-cell lymphoma-2 (Bcl-2) and the ratio of Bcl-2 to Bax in the jejunal mucosa of piglets challenged with PQ (*p* < 0.05) (Figure 4C–G). These data suggested that dietary silybin inhibited intestinal apoptotic procedures induced by the PQ challenge.

### 3.6. Dietary Silybin Addition Ameliorated PQ-Induced Intestinal Barrier Injury (Trial 2)

As shown in Figure 5A, PQ challenge significantly decreased jejunal villus height (VH) and the ratio of VH/CD (crypt depth) compared to those in the control (*p* < 0.05), but these negative changes were extensively restored with silybin supplementation (Figure 5A,B). The AB-PAS staining showed that the number of goblet cells in the jejunum tended to increase with dietary silybin supplementation (*p* = 0.07), thereby improving the content of MUC2 protein in the intestine of PQ-challenged piglets (Figure 5C–F). In addition, the activity of DAO in the plasma of PQ-challenged piglets was significantly increased (*p* < 0.05). However, dietary silybin administration significantly decreased DAO activity (*p* < 0.05) and tended to reduce the content of D-lactate (*p* = 0.05) (Figure 5G,H). Western blotting results showed that PQ challenge tended to reduce the protein expression of Occludin (*p* = 0.08), while silybin administration greatly enhanced the protein expression of ZO-1 under PQ challenge (*p* < 0.01). Also, the abundance of Occludin protein was higher in the Si plus PQ group than in the PQ group; still, it did not reach a statistically significant difference (*p* > 0.05), and there was no distinct change concerning the Claudin1 protein among all groups (*p* > 0.05) (Figure 5I,J). Taken together, these results indicated that silybin supplementation alleviated intestinal barrier dysfunction induced by the PQ challenge.

### 3.7. Dietary Silybin Addition Improved PQ-Induced Intestinal Microbiota Disorder (Trial 2)

We examined the effect of dietary silybin on the composition of the gut microbiota during the PQ challenge using 16S rRNA sequencing. The results of the α-diversity analysis revealed that PQ markedly lowered the Ace index and Chao1 index (*p* < 0.05) and tended to decrease the Shannon index (*p* < 0.1). While dietary silybin addition partly restored α-diversity indices of gut bacterial, there was no difference in α diversity between the Si + PQ and Ctrl groups (*p* > 0.05) (Figure 6A–C). A total of 2175 ASVs were discovered from the cecal chyme of all four groups, with 227 ASVs shared by these groups. More importantly, compared with the PQ challenge group, dietary silybin addition had more unique ASVs (380 vs. 229) (Figure 6D). The principal coordinates analysis plot indicated that the gut microbiota in the PQ challenge group formed a distinct cluster away from that of the other three groups (Figure 6E).

The distribution of the gut microbiota in cecal content was analyzed at the phylum and genus levels. For all groups, there was a dominant phylum dominance of Firmicutes and Bacteroidota in the intestinal microbiota (Figure 6F). Especially, the cecal microbiota composition was different at the genus level among the four groups (Figure 6G), and *Clostridium_sensu_stricto_1* and *Prevotella* were the two most abundant genera. Linear discriminant analysis and effect size analysis distinguished further series of differential species at different taxonomic levels (Appendix A). More precisely, Wilcoxon rank sum test analysis results indicated that the relative abundance of *Phascolarctobacterium* and *g__norank_f__T3* was evidently decreased in the PQ challenge group as compared to the control group (*p* < 0.05). Also, the PQ challenge tended to reduce the relative abundances of *Roseburia* (*p* = 0.09) and *Anaerovibrio* (*p* = 0.07) (Figure 6H). It is important to note that the relative proportion of *Prevotella* and *Phascolarctobacterium* drastically increased in response to dietary silybin treatment under the PQ challenge (*p* < 0.05). Meanwhile, silybin administration tended to increase the relative abundance of *Subdoligranulum* (*p* = 0.09) and *Roseburia* (*p* = 0.09), while a downward trend in *Bacteroides* (*p* = 0.09) was observed after silybin supplementation compared to the PQ challenge group (Figure 6I). Interestingly, several low-abundance genera, including *Dorea* and *g__norank_f__Lachnospiraceae*, were enriched by the PQ challenge, and the genus *Anaerovibrio*, *Olsenella*, and *g__unclassified_f__Butyricicoccaceae* were enriched in the Si + PQ group (*p* < 0.05) (Appendix A).

To probe the alterations in specific metabolites induced by the gut microbe remodeled with silybin, the concentration of SCFAs in the cecal digesta of piglets was examined. As expected, comparing the PQ challenge group with the control, a significant decrease in the concentration of acetate (*p* < 0.01) and a downward trend in propionate content (*p* = 0.07) were observed, while dietary silybin addition obviously increased the level of acetate and propionate in the cecum of PQ-challenged piglets (*p* < 0.05) (Figure 6J). Accordingly, the concentration of total SCFAs in the Si plus PQ group was significantly increased (*p* < 0.01) (Figure 6K). Additionally, there was no noticeable change concerning the content of formate, butyrate, isovalerate, and valerate among all groups (*p* > 0.05). These results suggested that dietary silybin improved intestinal microbiota composition and its metabolites in PQ-challenged piglets.

### 3.8. The Intestinal Microbiota and SCFAs Were Associated with Intestinal Homeostasis (Trial 2)

As shown in Figure 7A, there were significant correlations between the abundance of differential bacterial and intestinal homeostasis-related indexes (antioxidant capacity, mitochondrial function, anti-apoptotic, and intestinal function). In particular, the abundance of *Phascolarctobacterium*, *Roseburia*, and *Anaerovibrio* showed significant positive correlations with the expression of the Nrf2 gene and negative correlations with MDA content. The abundance of *Roseburia* was positively correlated with the activities of mitochondrial complex I, complex V, and ATP content. However, the abundance of *Phascolarctobacterium* showed significant negative correlations with the activities of caspase 3, caspase 9, and the expression of Cleaved caspase-3 protein, but positive correlations with the expression of Bcl-2 protein. In addition, the abundance of *Phascolarctobacterium*, *Roseburia*, and *Anaerovibrio* showed significant positive correlations with the expression of the ZO-1 protein. Additionally, we also found a direct correlation between the bacterial populations, such as the abundance of *Prevotella* being positively correlated with the abundance of *Phascolarctobacterium* and *Subdoligranulum*. More importantly, there were direct correlations between the abundance of some genera (such as *Prevotella*, *Phascolarctobacterium*, and *Subdoligranulum*) and SCFA contents (such as propionate and butyrate). Whilst, SCFAs were also tightly associated with intestinal homeostasis-related parameters, including antioxidant capacity, mitochondrial function, intestinal morphology, and the intestinal barrier (Figure 7A), redundancy analysis (RDA) results showed that the piglets that were fed a silybin-supplemented diet were aggregated near the acetate, propionate, and total SCFAs (Figure 7B).

## 4. Discussion

Silymarin has been used as a natural nutritional supplement to improve the growth performance and systemic health of some animals due to its diverse biological functions, such as antioxidant, anti-inflammatory, and regulation of lipid metabolism [22,23]. In the current study, we found for the first time that dietary silybin, the main active molecule of silymarin, can improve the growth performance by decreasing the FCR of weaned piglets, which is similar to the previous study in finishing pigs and broiler chicken [24,25]. Concurrently, another interesting finding in the present study is that supplementation with silybin could decrease diarrhea incidence, enhance CAT activity, and decrease MDA content in plasma, which indicates dietary silybin may improve the antioxidant capacity of weaned piglets and have beneficial effects on their intestinal health. Hence, we employed a model of intestinal injury induced via PQ to mimic the oxidative stress that piglets suffer from during weaning to explore the underlying mechanism of the health-improving effect of dietary silybin in weaned piglets [16].

As expected, our data showed that the PQ challenge significantly decreased GSH-Px activity and increased MDA content in the plasma of weaned piglets, which is consistent with previous reports [16]. This proposed that PQ triggered a strong systemic redox imbalance. It is important to maintain the proper redox homeostasis of the gastrointestinal tract as a prerequisite for intestinal health. The antioxidant enzymes secreted by intestinal epithelial cells are the first line of defense against redox imbalance [26]. In the present study, dietary silybin increased the CAT and SOD activities in the jejunum as well as CAT and GSH-Px in the plasma of PQ-challenged piglets and, as a result, reduced the concentration of MDA, a typical marker of lipid peroxidation, indicating that silybin addition alleviated PQ challenge-induced oxidative stress by augmenting antioxidant enzyme activities and subsequently decreased lipid peroxidation. Consistent with our results, it has been reported that silymarin supplementation alleviated oxidative stress caused by pregnancy by enhancing serum activities of CAT and GSH-Px in sows [11]. In addition, earlier studies have also shown that pretreatment with silymarin remarkably increased the activities of antioxidant enzymes in both in vivo and in vitro models [27,28]. It is well known that the Nrf2/Keap1 system plays a pivotal role in responding to oxidative stress. Upon oxidative stimulus, Nrf2 is released from its cytoplasmic repressor, Keap1, and then translocates to the nucleus to induce target gene expression [29]. In this study, our results showed that silybin restored *Nrf2* gene expression, which, in turn, upregulated the mRNA abundance related to antioxidant enzymes in the jejunum of PQ-challenged piglets. An in vitro study also found that silybin addition attenuated H_2_O_2_-induced oxidative stress and genotoxicity by modulating Nrf2 signaling [30]. Altogether, the results of this study signified that dietary silybin may resist the local oxidative stress of the intestinal tract by regulating the Nrf2 signaling pathway in PQ-challenged piglets.

As a critical organelle in the intestinal epithelial cell, the dynamic disorder and dysfunction of mitochondria are tightly involved in oxidative stress-related intestinal diseases [31]. The maintenance of mitochondrial dynamics depends heavily on proper biogenesis processes, including fission and fusion. The current study showed that the PQ challenge restrained mitochondrial division by downregulating the expression of *DNM1* and fusion by reducing the mRNA abundance of *MFN2*. These negative phenomena were partially restored with dietary silybin, which demonstrated that silybin supplementation improved mitochondrial biogenesis in the jejunum of PQ-challenged piglets. These results corroborate the previous study in which silybin alleviated ethanol- or acetaldehyde-induced injury in HepG2 cells, involving the reduction in DNM1 expression and the increase in MFN1 expression [32]. Also, another study found that silybin exerted antioxidant effects by regulating mitochondrial biogenesis in the liver of rats [33]. Mitochondria are counted as the factory of the cell, which generates ATP through oxidative phosphorylation relying on the respiratory chain, which is composed of five enzymatic complexes, including NADH dehydrogenase (complex I), succinate dehydrogenase (complex II), ubiquinol cytochrome c oxidoreductase (complex III), etc. [34]. A previous study showed that the PQ challenge causes protein oxidation in dopaminergic neurons by interfering with mitochondrial complex I activity [35]. Herein, dietary silybin upregulated *NDUFV2* (Complex I) and *ATP5H* (Complex V) gene expression in the jejunum of PQ-challenged piglets. Consistently, the activities of mitochondrial complex I and complex V were tremendously improved with silybin addition, thereby increasing ATP levels in the jejunum of piglets challenged with PQ. Actually, our results were in support of previous research where silybin prevented cardiolipin peroxidation by recovering complex I and V activity [33]. Interestingly, correlation analysis showed that mitochondrial function-related genes and the activities of the mitochondrial complex were positively correlated with the expression of Nrf2 signaling pathway genes (Figure 3K and Figure 7A), indicating that silybin supplementation may enhance intestinal antioxidant capacity by regulating mitochondrial function to cope with the oxidative stress induced by PQ challenge in piglets.

Mitochondrial dysfunction would trigger endogenous apoptotic procedures, the initiation of which is mainly manifested via the activation of caspase 9, which can activate caspase 3, a major executioner of the apoptotic procedure in mammals [36]. Previous studies suggested that the PQ challenge enhanced caspase 9 and caspase 3 activities and induced apoptotic procedures [37]. Consistently, the current study confirmed this result, while dietary silybin supplementation effectively prevented the increase in caspase 9 activity, thereby decreasing the activity of caspase 3, which indicated that silybin inhibited intestinal apoptotic procedures induced via PQ in weaned piglets. Previous studies also reported that silymarin or silybin played an anti-apoptotic effect by preventing the activation of caspases [38,39]. It is well known that the endogenous apoptotic pathway is regulated by pro-apoptotic members (such as Bax) and anti-apoptotic members (such as Bcl-2) of the Bcl-2 protein family [37]. Bcl-2 inhibits apoptotic procedures by controlling mitochondrial permeability and cytochrome C release [40]. PQ decreased the mRNA expression level of Bcl-2 and resulted in liver cell apoptosis and death in mice [41]. Herein, we found that the PQ challenge reduced the expression level of Bcl-2 protein and the ratio of Bcl-2 to Bax, while a low ratio of Bcl-2 to Bax implied increased activation of caspases, which was consistent with increased protein expression of Cleaved caspase-3 caused by the PQ challenge. Conversely, dietary silybin treatment greatly reversed these aberrant changes, which may be related to the relief of mitochondrial damage. A previous study also suggested silymarin attenuated paraquat-induced apoptosis in macrophages by recovering Bcl-2 protein expression and reducing Bax protein expression [27]. Altogether, these results confirmed that dietary silybin mitigated PQ-induced intestinal cell apoptotic procedures by regulating mitochondrial-mediated endogenous apoptotic pathways.

The intestinal barrier plays a vital role in preventing harmful substances and pathogenic bacteria from entering the circulatory system, which is composed primarily of a mechanical barrier, a chemical barrier, and a biological barrier [42]. The present study showed that the PQ challenge caused intestinal wall damage, decreased villus height and the ratio of VH to CD in the jejunum, indicating an adverse impact on the intestinal morphology and integrity of piglets. Whereas these negative changes were greatly attenuated with silybin addition in piglets, which may explain why there was an improvement in the growth arrest with dietary silybin in this current study. Other studies also reported similar results [43]. MUC2 is one of the most abundant mucins released in the intestinal lumen via goblet cells and the key to constructing a sturdy mucus layer [44], which is not only a physical barrier to isolate pathogenic bacteria but also provides nutrition and adhesion sites for symbiotic bacteria in the intestine [45]. Herein, we found that dietary silybin increased the number of goblet cells and subsequently resulted in an increased content of MUC2 in the jejunum, indicating a beneficial role of dietary silybin in improving the chemical barrier. Likewise, analogous findings were obtained for dietary silymarin [46]. The mechanical barrier is the most critical intestinal mucosal barrier, the structural basis of which is mainly the enterocytes and the tight junction between epithelial cells, while ZO-1, Occludin, and Claudin1 are the primary components of tight junction proteins [47]. In this study, dietary silybin administration enhanced the expression of ZO-1 and Occludin protein in the jejunum of PQ-challenged piglets. This result may explain why there was a decrease in plasma DAO activity and D-lactate levels with dietary silybin. Collectively, these data demonstrated that silybin supplementation alleviated PQ-induced intestinal injury by improving intestinal barrier function. As far as we know, this is the first study to confirm the beneficial effect of silybin on the intestinal barrier function of piglets.

Accumulating evidence has indicated the pivotal role of gut microbiota in the intestinal health of mammals and helped repair intestinal mucosal barrier damage [48]. In this study, we found that dietary silybin partly restored the decline in cecal microbiota diversity induced by the PQ challenge and obtained more unique ASV. In detail, silybin addition altered microbiota structure and interactions at the genus level. Thereinto, higher abundances of *Prevotella* and lower abundances of *Bacteroides* were observed in PQ-challenged piglets after supplementing dietary silybin. A previous study showed that *Prevotella* might improve the absorption of monosaccharides and bring performance advantages to the host [49]. *Bacteroides* is an opportunistic pathogen that can induce colitis [50,51]. More importantly, our study found dietary silybin increased the abundance of *Phascolarctobacterium*, and it can produce acetate and propionate by consuming succinate and pyruvate [52]. It is well known that succinate in the intestinal cavity is essential for the growth of *Clostridioides difficile*, and Blooming *C. difficile* causes life-threatening intestinal inflammation [53]. Similar to our results, previous studies also found that silybin could improve intestinal microbial ecology by upregulating the relative abundance of *Akkermansia* and *Allobaculum* in Alzheimer’s disease mice and inhibiting the proliferation of ileal pathogenic bacteria [13,54]. Moreover, silybin addition also increased the proportion of *Subdoligranulum*, *Roseburia*, *Anaerovibrio,* and *Olsenella*. It has been reported that these genera have the ability to produce SCFAs [49,55,56,57]. Correspondingly, we also found dietary silybin increased the content of acetate, propionate, and total SCFAs in cecal digesta. Growing evidence has demonstrated that SCFAs assist in maintaining gut barrier function by regulating intestinal epithelial integrity and redox balance, which contribute to preventing the development of intestinal diseases [58]. Consistently, correlation and RDA analyses further indicated that SCFAs exhibited a high degree of consistency and correlation with intestinal homeostasis-related parameters, including antioxidant capacity, mitochondrial function, intestinal barrier, and gut microbiota (Figure 7A,B). Overall, this study showed that gut microbe and SCFA alterations responding to dietary silybin administration presumably are crucial contributors to the maintenance of the intestinal health of PQ-challenged piglets.

## 5. Conclusions

In conclusion, our results suggested that dietary silybin supplementation linearly decreased feed conversion ratio and diarrhea incidence in normal feeding conditions. In particular, silybin supplementation could enhance mitochondrial function and improve the intestinal barrier and microflora community in response to PQ-induced oxidative damage. These findings imply that dietary silybin supplementation may constitute an attractive strategy for improving intestinal health in juvenile animals, which also provides theoretical support for the development and application of silybin as a natural feed additive.

## Figures and Tables

**Figure 1 antioxidants-12-01975-f001:**
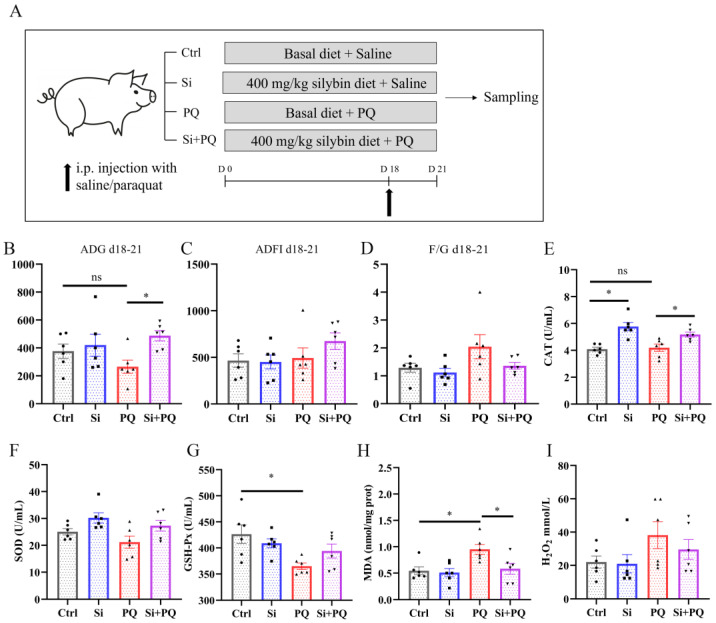
Dietary silybin supplementation alleviated the redox imbalance and growth retardation induced by paraquat challenges in piglets. (**A**) Schematic diagram of experimental design. (**B**) ADG on days 18–21. (**C**) ADFI on days 18–21. (**D**) F/G on days 18–21. (**E**–**I**) The activity of plasma CAT (**E**), SOD (**F**), GSH-Px (**G**), and the concentration of MDA (**H**) and H_2_O_2_ (**I**). Data are expressed as mean ± standard error. * *p* < 0.05. Ctrl = basal diet group treated with saline; Si = silybin diet group treated with saline; PQ = basal diet group treated with paraquat; Si + PQ = silybin diet group treated with paraquat; ns = not significant; ADG = average daily gain; ADFI = average daily feed intake; F/G = ADFI/ADG; CAT = catalase; SOD = superoxide dismutase; GSH-Px = glutathione peroxidase; MDA = malondialdehyde; and H_2_O_2_ = hydrogen peroxide.

**Figure 2 antioxidants-12-01975-f002:**
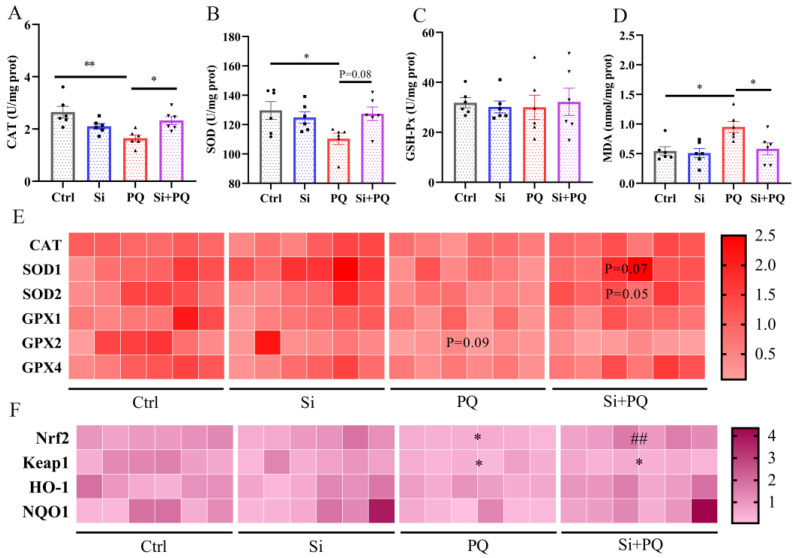
Dietary silybin supplementation alleviated paraquat-induced intestinal oxidative stress in piglets. The activities of CAT (**A**), SOD (**B**), GSH-Px (**C**), and the level of MDA (**D**) in the jejunum. The heat maps of the mRNA abundance for antioxidant enzyme genes (**E**) and Nrf2/Keap1 signaling pathway genes (**F**), * *p* < 0.05 vs. Ctrl group; ## *p* < 0.01 vs. PQ group. Data are expressed as mean ± standard error. * *p* < 0.05 and ** *p* < 0.01. Ctrl = basal diet group treated with saline; Si = silybin diet group treated with saline; PQ = basal diet group treated with paraquat; Si + PQ = silybin diet group treated with paraquat; CAT = catalase; SOD = superoxide dismutase; GSH-Px = glutathione peroxidase; MDA = malondialdehyde; *GPX* = glutathione peroxidase; *Nrf2* = nuclear factor-erythroid 2-related factor 2; *Keap1* = kelch-like ECH-associated protein l; *HO-1* = heme oxygenase-1; and *NQO1* = NAD(P)H: quinone oxidoreductase 1.

**Figure 3 antioxidants-12-01975-f003:**
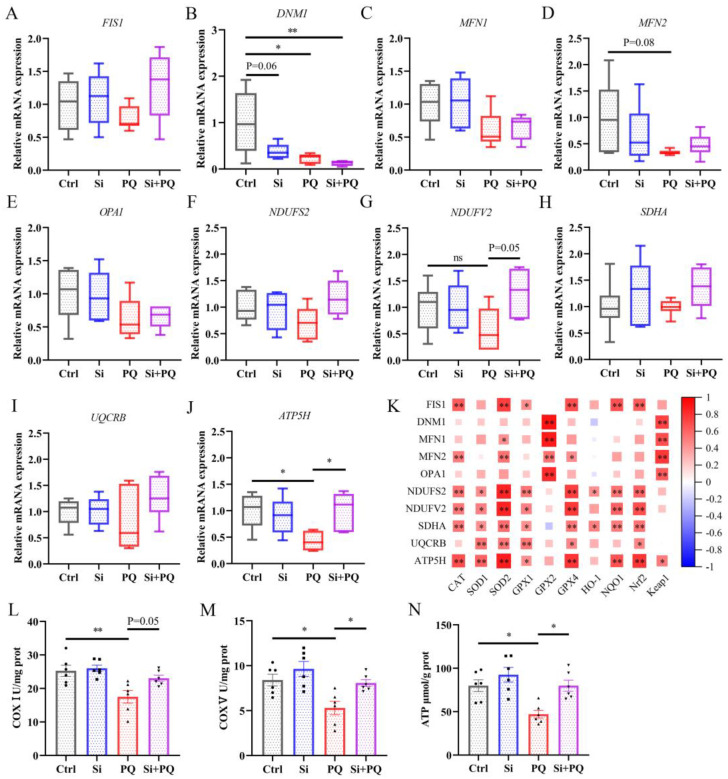
Dietary silybin supplementation protected against PQ-induced mitochondrial injury. (**A**–**E**) The expression of mitochondrial biogenesis genes includes *FIS1* (**A**), *DNM1* (**B**), *MFN1* (**C**), *MFN2* (**D**), and *OPA1* (**E**). The relative mRNA abundance of mitochondrial respiratory chain membrane protein-related genes includes *NDUFS2* (**F**), *NDUFV2* (**G**), *SDHA* (**H**), *UQCRB* (**I**), and *ATP5H* (**J**). The heatmap of Spearman’s correlation between the expression of mitochondrial function-related genes and Nrf2 pathway genes (**K**). The activities of mitochondrial complex I (**L**), complex V (**M**), and ATP level (**N**) in the jejunum. Data are expressed as mean ± standard error. * *p* < 0.05 and ** *p* < 0.01. Ctrl = basal diet group treated with saline; Si = silybin diet group treated with saline; PQ = basal diet group treated with paraquat; Si + PQ = silybin diet group treated with paraquat; *FIS1* = fission; *DNM1* = dynamin 1; *MFN* = mitofusin; *OPA1* = mitochondrial dynamin-like GTPase; *NDUFS2* = NADH ubiquinone oxidoreductase core subunit S2; *NDUFV2* = NADH ubiquinone oxidoreductase core subunit V2; *SDHA* = succinate dehydrogenase complex flavoprotein subunit A; *UQCRB* = ubiquinol-cytochrome c reductase binding protein; *ATP5H* = ATP synthase subunit d; *CAT* = catalase; SOD = superoxide dismutase; *GPX* = glutathione peroxidase; *Nrf2* = nuclear factor-erythroid 2-related factor 2; *Keap1* = kelch-like ECH-associated protein l; *HO-1* = heme oxygenase-1; *NQO1* = NAD(P)H: quinone oxidoreductase 1; COX = mitochondrial complex; and ATP = adenosine triphosphate.

**Figure 4 antioxidants-12-01975-f004:**
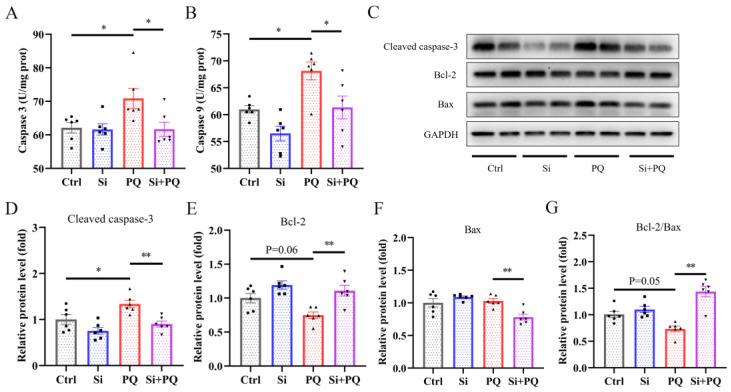
Dietary silybin addition mitigated PQ-induced intestinal apoptotic procedures. The activities of caspase 3 (**A**) and caspase 3 (**B**) in the jejunum. (**C**–**G**) The levels of apoptotic proteins in the jejunum were detected using Western blot. Data are expressed as mean ± standard error. * *p* < 0.05 and ** *p* < 0.01. Ctrl = basal diet group treated with saline; Si = silybin diet group treated with saline; PQ = basal diet group treated with paraquat; Si + PQ = silybin diet group treated with paraquat; Bcl-2 = B-cell lymphoma-2; Bax = Bcl-2-associated-X-protein; and Bcl-2/Bax = the ratio of Bcl-2 to Bax.

**Figure 5 antioxidants-12-01975-f005:**
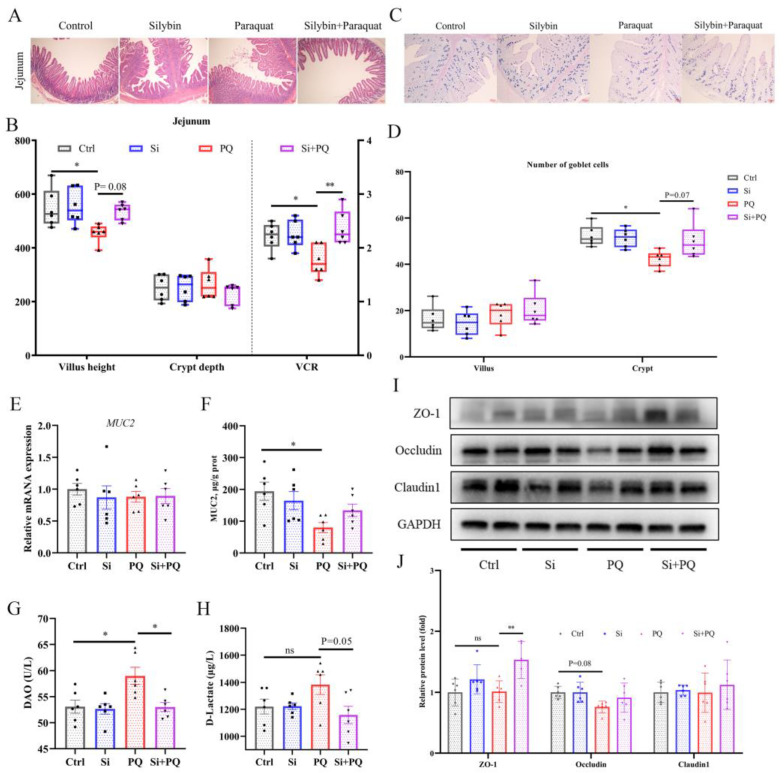
Dietary silybin addition ameliorated PQ-induced intestinal barrier dysfunction. (**A**) Representative images of H&E staining of the jejunum sections (scale bars: 40 μm). Quantification of intestinal morphology of the jejunum (**B**). Goblet cells in the jejunum were observed via AB-PAS staining (scale bars: 100 μm) (**C**) and quantification (**D**). The expression of MUC2 at the gene level (**E**) and protein level (**F**). (**G**,**H**) Plasma DAO activity and D-lactate content. (**I**,**J**) The levels of tight junction proteins in the jejunum were detected with Western blot. Data are expressed as mean ± standard error. * *p* < 0.05 and ** *p* < 0.01. Ctrl = basal diet group treated with saline; Si = silybin diet group treated with saline; PQ = basal diet group treated with paraquat; Si + PQ = silybin diet group treated with paraquat; VCR = the ratio of villus height-to-crypt depth; MUC2 = mucin 2; DAO = diamine oxidase; and ZO-1 = zonula occludens-1.

**Figure 6 antioxidants-12-01975-f006:**
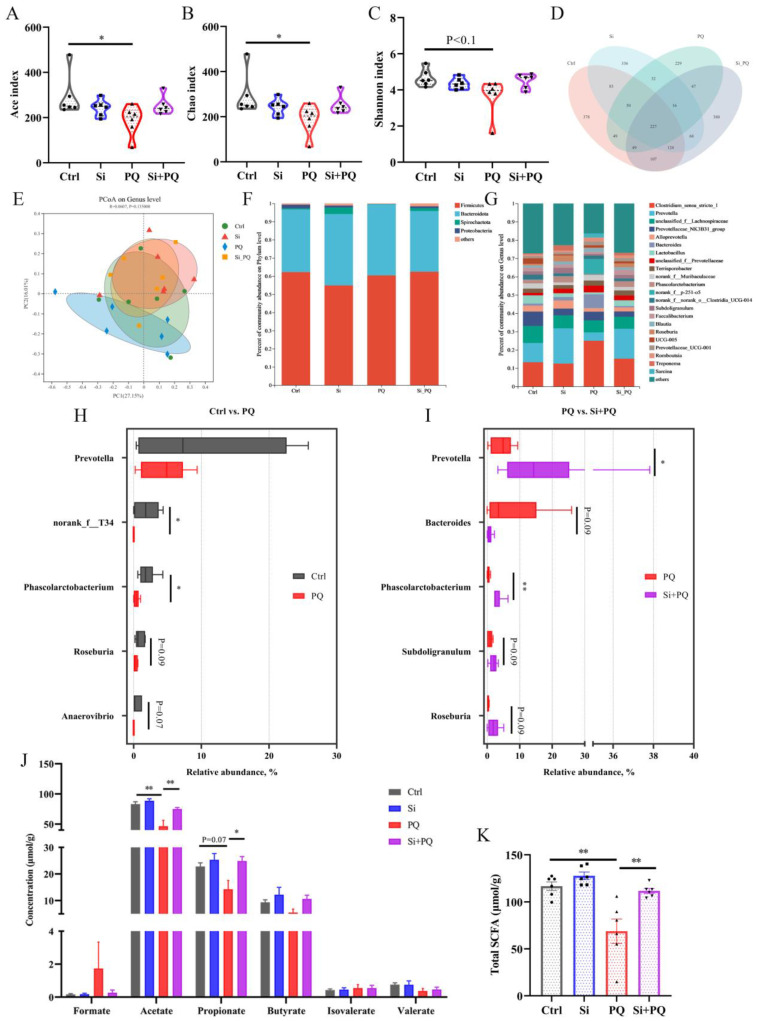
Dietary silybin addition improved PQ-induced intestinal microbiota disorders. (**A**–**C**) α-diversity index. (**D**) Ven diagram of ASV in cecal chyme of all samples. (**E**) Principal coordinate analysis. The relative abundance of gut microbiota at the phylum (**F**) and genus level (**G**). (**H**,**I**) Gut microbiota with significant differences of the comparison groups Ctrl vs. PQ and PQ vs. Si + PQ at the genus levels. (**J**,**K**) The concentration of cecal short-chain fatty acids. Data are expressed as mean ± standard error. * *p* < 0.05 and ** *p* < 0.01. Ctrl = basal diet group treated with saline; Si = silybin diet group treated with saline; PQ = basal diet group treated with paraquat; Si + PQ = silybin diet group treated with paraquat; PCoA = principal coordinates analysis; and SCFAs = short-chain fatty acids.

**Figure 7 antioxidants-12-01975-f007:**
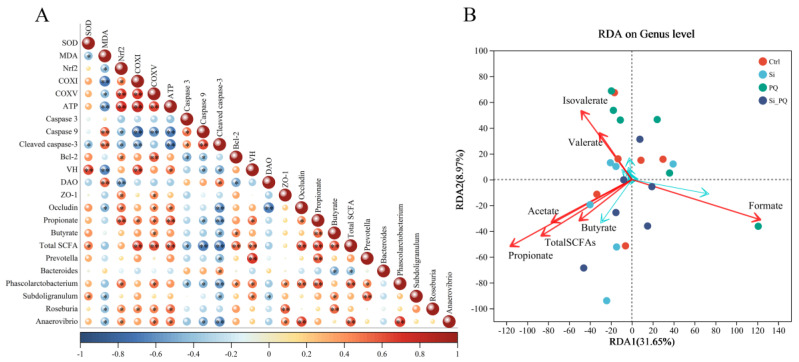
Spearman correlation analysis and RDA analysis. (**A**) The heatmap of Spearman’s correlation between the abundance of gut microbiota and the intestinal homeostasis phenotype indexes. (**B**) The RDA of the differences between the gut microbiota and SCFA levels. * *p* < 0.05 and ** *p* < 0.01. CAT = catalase; SOD = superoxide dismutase; MDA = malondialdehyde; Nrf2 = nuclear factor-erythroid 2-related factor 2; COX = mitochondrial complex; ATP = adenosine triphosphate; ATP5H = ATP synthase subunit d; VH = villus height; V/C = the ratio of villus height-to-crypt depth; DAO = diamine oxidase; ZO-1 = zonula occludens-1; SCFAs = short-chain fatty acids; and RDA = redundancy analysis.

**Table 1 antioxidants-12-01975-t001:** Effects of dietary silybin supplementation on growth performance and diarrhea incidence in weaned piglets (trial 1) ^1^.

Item	Silybin Level, mg/kg					SEM	*p*-Value		
	0	50	100	200	400		ANOVA	Linear	Quadratic
BW, kg									
Day 0	8.28	8.28	8.28	8.28	8.28	0.49	1.00	1.00	1.00
Day 14	10.39	10.60	10.57	10.23	11.00	0.53	0.88	0.51	0.59
Day 28	16.05	16.05	15.85	16.18	16.51	0.81	0.99	0.62	0.84
Day 42	23.33	23.42	23.63	24.41	24.80	1.12	0.86	0.28	0.88
ADG ^2^, g									
Day 0–14	151	166	164	140	194	16.76	0.25	0.15	0.23
Day 14–28	404	389	377	422	394	26.05	0.82	0.92	0.86
Day 28–42	520	526	556	588	592	25.53	0.20	0.03	0.34
Day 0–42	358	360	366	388	393	17.56	0.54	0.11	0.73
ADFI ^3^, g									
Day 0–14	270	297	280	262	316	22.26	0.47	0.29	0.36
Day 14–28	659	674	671	700	692	29.66	0.90	0.43	0.62
Day 28–42	996	927	963	1038	977	38.63	0.46	0.68	0.61
Day 0–42	642	633	638	672	662	26.63	0.86	0.42	0.75
FCR ^4^									
Day 0–14	1.81	1.86	1.72	1.96	1.65	0.11	0.30	0.34	0.23
Day 14–28	1.67	1.74	1.79	1.68	1.78	0.08	0.77	0.53	0.89
Day 28–42	1.92 ^a,x^	1.77 ^ab^	1.74 ^ab,y^	1.77 ^ab^	1.65 ^b^	0.05	0.02	0.00	0.35
Day 0–42	1.80	1.76	1.75	1.74	1.68	0.03	0.24	0.03	0.91
DI ^5^, %									
Day 0–14	8.93 ^a^	5.36 ^ab^	6.55 ^a^	7.32 ^a^	2.98 ^b^	-	0.02	-	-

SEM = standard error of the mean; BW = body weight. ^1^ Six replicates per treatment (*n* = 6), four piglets per replicate; the total number of animals is 120. ^2^ Average daily gain (ADG) = body weight gain of the pen/piglets’ number/days. ^3^ Average daily feed intake (ADFI) = feed intake of the pen/piglets’ number/days. ^4^ Feed conversion ratio (FCR) = feed intake of the pen/body weight gain of the pen. ^5^ Diarrhea incidence (DI), % = the number of piglets with diarrhea/the total number of weaned piglets × 100%. ^a,b^ Values within a row without common letters differ significantly (*p* < 0.05). ^x,y^ Values listed in the same row with different superscripts tend to be different (0.05 ≤ *p* < 0.1).

**Table 2 antioxidants-12-01975-t002:** Effects of dietary silybin supplementation on plasma antioxidant capacity in weaned piglets (trial 1) ^1^.

Item	Silybin Level, mg/kg					SEM	*p*-Value		
	0	50	100	200	400		ANOVA	Linear	Quadratic
Day 14									
CAT, U/mL	2.14	2.47	1.93	2.08	2.05	0.23	0.55	0.51	0.71
SOD, U/mL	29.23 ^b^	35.77 ^a^	34.53 ^a^	34.11 ^a^	35.19 ^a^	0.78	0.00	0.01	0.01
GSH-Px, U/mL	380	411	373	374	394	15.3	0.39	0.93	0.41
MDA, nmol/mL	2.82 ^x^	2.82	2.29	2.23 ^y^	2.39	0.14	0.03	0.03	0.02
Day 42									
CAT, U/mL	3.88 ^b^	4.08 ^ab^	4.23 ^ab^	4.39 ^ab^	4.7 ^a^	0.18	0.05	0.00	0.56
SOD, U/mL	32.56	30.84	32.68	31.13	35.04	1.08	0.16	0.08	0.15
GSH-Px, U/mL	374	366	352	401	399	13.3	0.08	0.04	0.95
MDA, nmol/mL	2.36 ^a,x^	2.26 ^ab^	2.21 ^ab^	1.83 ^ab,y^	1.73 ^b^	0.14	0.01	0.00	0.35

SEM = standard error of the mean; CAT = catalase; SOD = superoxide dismutase; GSH-Px = glutathione peroxidase; and MDA = malondialdehyde. ^1^ Six replicates per treatment (*n* = 6). ^a,b^ Values within a row without common letters differ significantly (*p* < 0.05). ^x,y^ Values listed in the same row with different superscripts tend to be different (0.05 ≤ *p* < 0.1).

## Data Availability

Data is contained within the article and Supplementary Material.

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
