# Peer review of "The Effects of Dietary Silybin Supplementation on the Growth Performance and Regulation of Intestinal Oxidative Injury and Microflora Dysbiosis in Weaned Piglets"

_antioxidants, 2023, doi:10.3390/antiox12111975_

Round 1
Reviewer 1 Report
Comments and Suggestions for Authors
Author studied dietary silybin supplementation on the growth performance and regulating intestinal oxidative injury and microflora dysbiosis in weaned piglets.
Line 89 Please change to two male and two female. Was male pig castrated? If it was, then change it barrow.
Line 90 How was the level of inclusion rate determined?
Line 97 is it cost effectively to consider included the product at 400 mg/kg in diet? How many male and female pigs were used?
Line 107 Please remove “The ingredient and”
Line 108 Please double check the NRC 2012. I believe the dietary amino acids are lower than the levels that are suggested by NRC.
Line 114 was pigs weighted individually or per pen basis?
Line 124 What was gender distribution for each treatment?
Line 130 What is longitudinally?
Line 251 While treatments that contained Silybin had close results on FCR when calculated from ADFI and ADG, control treatment appeared to off. Please double check the number.
Line 274 Missing the superscript on MDA
Comments on the Quality of English Language
Revised terminologies are required.
Reviewer 2 Report
Comments and Suggestions for Authors
Major comments
In abstract, which are the exact beneficial effects on growth performance, diarrhea, and plasma antioxidant capacity indices? Overall, we recommend to provide less details in abstract, as it is too excessive.
Introduction: Add more relevant studies in pig models.
In Tables, please explain how the superscripts were added. For example, in Table 2, in MDA day 14, the superscripts x,y indicate tendency, however all p-values given are p<0.05. Add superscripts if they are missing.
It would be useful for the reader to mention which methods, results and Tables refer to piglets of trial 1 or trial 2. It is clarified in experimental design, but we recommend to remind these information at the other parts of the manuscript.
In Results, there are some sentences that provide explanations, and therefore they might be better placed in Discussion section. Examples: L287, 337, 370, 393.
In Conclusion, L647-649, clarify that these are findings of the study, as they are presented only as possible mechanisms.
Minor comments
L54-55: Explain what is A/B.
L64-66: Add information about the experimental model.
L28 and elsewhere: replace “apoptotic” with “apoptotic procedures”.
L66: Replace “the beneficial effects” with “any beneficial effects” as this is hypothetical.
L77: Add the word “dose” or “dosage” next to “dietary proper silybin”.
L79: “Improving intestinal health” should be written as a possible effect and not a statement.
L104-106: The sentence “which suggested…group” describes results, so it should not be placed in Materials and Methods.
L126: Rephrase.
L138: H2O2, the number 2 should be written as an index.
L230: Correct “honey” (replace with “honest”).
L292: Better use the word “alleviated”.
In Figures, add the silybin dosage and avoid using verbs.
L646-647: “Growth performance parameters” is more accurate.
Reviewer 3 Report
Comments and Suggestions for Authors
The authors must change conclusion part as this does not refer to their exact findings. The authors did at first a study with control and 4 diets with silybin. Although, they find no difference on performance, they decided to test further the highest dose. Why?
First, they should do one round of several antioxidant parameters.
Also, table 1, does not have letters as index for FCR.
Paraquat mostly affects the respiratory tract and not the gastrointestinal tract. Maybe, the reason of diarrhea is different than that the authors believe.
Photos in Figure 4 A, do not have any difference on different groups.
Intestinal barrier and histology photos also seem to have no difference among the groups
Round 2
Reviewer 2 Report
Comments and Suggestions for Authors
authors have implemented major changes in the manuscript
Reviewer 3 Report
Comments and Suggestions for Authors
Authors responded and revised their work in a very satisfactory way. All points raised- as in my point of view- gave a confusing conclusion- now have been revised that the results and conclusions are clear and figures show the appropriate and relevant results.